# Spectroscopy and dynamics of the hydrated electron at the water/air interface

Caleb J. C. Jordan[1], Marc P. Coons [2], John M. Herbert [2] ✉ & Jan R. R. Verlet [1] ✉

The hydrated electron, $e^-_{(aq)}$, has attracted much attention as a central species in radiation chemistry. However, much less is known about $e^-_{(aq)}$ at the water/air surface, despite its fundamental role in electron transfer processes at interfaces. Using time-resolved electronic sum-frequency generation spectroscopy, the electronic spectrum of $e^-_{(aq)}$ at the water/air interface and its dynamics are measured here, following photo-oxidation of the phenoxide anion. The spectral maximum agrees with that for bulk $e^-_{(aq)}$ and shows that the orbital density resides predominantly within the aqueous phase, in agreement with supporting calculations. In contrast, the chemistry of the interfacial hydrated electron differs from that in bulk water, with $e^-_{(aq)}$ diffusing into the bulk and leaving the phenoxyl radical at the surface. Our work resolves long-standing questions about $e^-_{(aq)}$ at the water/air interface and highlights its potential role in chemistry at the ubiquitous aqueous interface.

Much like classical anions, electrons can behave as solutes in solution. In water, such hydrated electrons ($e^-_{(aq)}$) have attracted much attention as fundamental quantum solutes and because of their role in radiation chemistry[1,2]. The structure and dynamics of $e^-_{(aq)}$ has been a topic of much debate[3], with key outstanding questions relating to solvation at the water/air interface[4]. Specifically, does the electron's charge distribution reside predominantly above or below the water surface, and how long does the electron at the water/air interface ($e^-_{(aq/air)}$) remain near the surface? These questions are pertinent because, in many instances, $e^-_{(aq)}$ is expected to be found at interfaces, with implications ranging from atmospheric, interstellar and radiation chemistry to quantum solvation, interfacial charge-transfer and plasma processes[1,5–9]. As a specific example, $e^-_{(aq/air)}$ has been implicated in the recently observed enhancement of reactivity in microdroplets, where the electron is assumed to diffuse rapidly into the bulk[10].

The consensus view of the structure of $e^-_{(aq)}$ is one where the electron density predominantly resides within a cavity or excluded volume in the water structure[3,11]. It can be conceptualized as an electron in a quasi-spherical box with an electronic ground state defined by a nodeless s-type orbital. Its first excited states are three p-type states and p ← s photo-excitation accounts for much of the optical absorption spectrum, which is the most characteristic observable of $e^-_{(aq)}$[3,12,13]. But how does this cavity structure change at the water/air interface? There have been conflicting views built upon photoelectron spectroscopy of water cluster anions, where experiments demonstrate the existence of differing binding motifs for the electron[14,15]. Some clusters correlate with embryonic forms of $e^-_{(aq)}$, where most of the electron distribution resides below the surface (inside the cluster) while other motifs are more weakly bound, consisting of a partially hydrated electron with most of its electron distribution protruding into the vapor phase[16,17]. Experiments on clusters deposited on cold metal surfaces found evidence for the latter[18], as did an early photoelectron spectroscopy experiment of a water microjet[19]. However, the signal attributed to interfacial $e^-_{(aq)}$ is much shorter lived in other microjet experiments[20], consistent with an excited state[3]. Recent heterodyne-detected vibrational sum-frequency generation (SFG) spectroscopy of the ambient water/air interface suggests a partially hydrated electron[21], but electronic second-order non-linear spectroscopy at specific wavelengths appears to show kinetics that are broadly consistent with those for $e^-_{(aq)}$ buried in the interface[22,23]. The theoretical consensus is that $e^-_{(aq/air)}$ has most of its electron density in the aqueous phase[24–26].

Significant experimental effort has been devoted to measuring the vertical detachment energy using photoelectron spectroscopy

[1]Department of Chemistry, Durham University, Durham DH1 4LJ, UK. [2]Department of Chemistry and Biochemistry, The Ohio State University, Columbus, OH, USA. ✉e-mail: herbert@chemistry.ohio-state.edu; j.r.r.verlet@durham.ac.uk

because this quantity can distinguish between the two binding motifs. Such measurements are not readily transferable to an ambient water/air interface, however. In experiments using liquid microjets, which are proxies for the ambient water/air interface, there have been contrasting results[19,20,27–32]. The electronic absorption spectrum, on the other hand, has been *the* defining experimental feature of e⁻$_{(aq)}$[33–36]. Its measurement at the ambient water/air interface has not been reported, although it is expected to be sensitive to surface localization[37,38]. Here, we use time-resolved electronic SFG spectroscopy to measure the spectrum and subsequent solvation dynamics of e⁻$_{(aq/air)}$, thereby directly addressing the two key outstanding questions related to the solvation of electrons at the water/air interface.

## Formation and spectroscopy of e⁻$_{(aq/air)}$

SFG relies on the second-order non-linear response of a material to an electromagnetic field[39–41]. In the electric dipole approximation, this response is only finite where centro-symmetry is broken, which is necessarily the case at the interface between isotropic phases such as water and air. Therefore, two driving fields with frequencies $\omega_1$ and $\omega_2$ will combine to generate the SFG field with frequency $\omega_{SFG} = \omega_1 + \omega_2$, exclusively from the interface. The field $\omega_{SFG}$ can be enhanced when any of the three fields ($\omega_1$, $\omega_2$ or $\omega_{SFG}$) are resonant with an optical transition of the interfacial species. In the present experiments, both e⁻$_{(aq)}$ and e⁻$_{(aq/air)}$ were generated by photo-excitation of phenoxide anions using a pump pulse $\omega_{pump}$ ($\lambda = 257$ nm)[42], which predominantly accesses the $S_1 \leftarrow S_0$ transition, leading to the formation of a fully solvated electron. The phenoxide anion is surface active[43] and serves as a prototypical moiety, participating for example in photo-oxidation of chromophores in the green fluorescent and photoactive yellow proteins[44]. The non-linear response was generated from a variable frequency field, $\omega_1$ ($\lambda = 620–800$ nm), and a fixed frequency field, $\omega_2$ ($\lambda = 1026$ nm), producing $\omega_{SFG}$ ($\lambda = 386–450$ nm). Both $\omega_1$ and $\omega_2$ were delayed together with respect to $\omega_{pump}$ to allow for time-resolved SFG spectroscopy, which is essential because of the transient nature of e⁻$_{(aq)}$. A schematic of the experiment is shown in Fig. 1a with further experimental details in the Methods section.

To obtain a spectrum, $\omega_1$ was scanned and a kinetic trace of the dynamics was measured up to $t = 60$ ps, at each wavelength. Specific consideration was given to experimental parameters between measurements at different $\omega_1$ to ensure that the relative signals measured were comparable (see Methods). In the limit of weak non-resonant signal from the nascent water/air interface, the measured SFG signal ($I_{SFG}$) depends quadratically on the surface concentration of absorbers when any of the $\omega_1$, $\omega_2$, or $\omega_{SFG}$ fields are resonant with a transition. As $\omega_1$ was scanned across a range of the absorption spectrum of e⁻$_{(aq)}$, resonance-enhancement at the interface may be anticipated as shown in Fig. 1b. UV excitation of phenoxide also produces the phenoxyl radical, PhO·[45–47]. The latter has an absorption spectrum peaking at $\lambda = 400$ nm (in aqueous solution) corresponding to the $C^2B_1 \leftarrow X^2B_1$ transition[48]. This transition coincides with the wavelength range of $\omega_{SFG}$ and, therefore, may also appear in the signal through resonance enhancement, as shown in Fig. 1b.

## Results

Figure 2a shows the square-root of the SFG signal, $I_{SFG}^{1/2}$ (proportional to interfacial concentration), as a function of time and over a range of $\omega_1$. Signal before $t = 0$ has been subtracted, residual fluorescence contributions removed, and traces offset for clarity. At all $\omega_1$, the SFG signal rises at $t = 0$ within the instrumental time-resolution (~200 fs) and then decays on a longer timescale. However, the decay kinetics are markedly different for differing $\omega_1$: as $\omega_1$ is changed to higher frequency (shorter wavelength), the traces appear to show an offset in signal at longer times ($t = 60$ ps) and a much smaller decaying contribution.

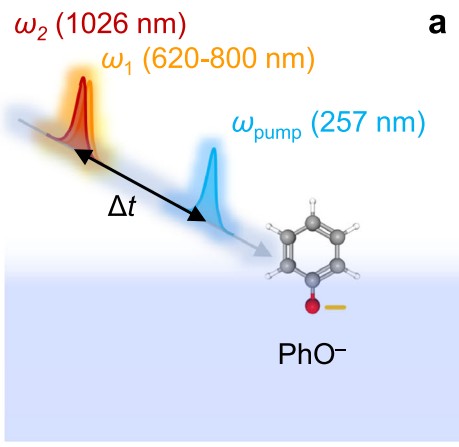

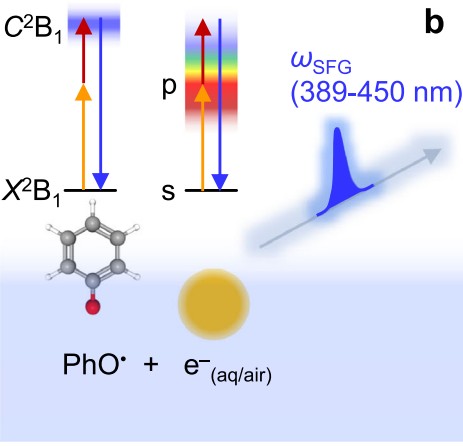

**Fig. 1 | Schematic of experiment and transitions. a** Excitation and probing scheme for phenoxide, PhO⁻, at the water/air interface. **b** Relevant transitions following photo-oxidation of phenoxide, leading to sum-frequency generation (SFG) signals produced by resonant enhancement of $\omega_1$ for the hydrated electron at the water/air interface, e⁻$_{(aq/air)}$, and of $\omega_{SFG}$ for the phenoxyl radical, PhO·.

The data in Fig. 2a were analyzed using a global fitting methodology to the total signal, $I_{SFG}(t, \lambda)^{1/2}$. A kinetic model involving two species ($i = A$ and B) is assumed, whose concentrations have simple first-order kinetics with lifetimes $\tau_i$:

$$I_{SFG}(t,\lambda)^{1/2} = \sum_i c_i(\lambda) \exp(-t/\tau_i) * G(t),$$

where * indicates convolution with a Gaussian instrument response function, $G(t)$, and $c_i(\lambda)$ are amplitudes that correspond to a spectrum which is associated with the decay constant of species $i$ (further details in Supplementary Note 1). Figure 2a includes the results of the fit, which accounts for all the observed dynamics with no clear systematic deviations, suggesting that the two-component model and assumption about the kinetics have captured the processes taking place. The two lifetimes obtained are $\tau_A = 12 \pm 1$ ps and $\tau_B > 100$ ps. The decay-associated spectra, $c_i(\lambda)$, are shown in Fig. 2b. These data reveal that the spectrum associated with species A, which decays with a lifetime $\tau_A$, peaks around $\lambda = 720$ nm, whereas that associated with B (decaying with a lifetime $\tau_B > 100$ ps) has a low amplitude at longer wavelengths and rises towards shorter wavelengths.

In Fig. 3, the absorption spectrum of the hydrated electron (at 298 K)[34] is shown along with the spectrum of interfacial species A. The spectrum associated with A has the general appearance of e⁻$_{(aq)}$ with

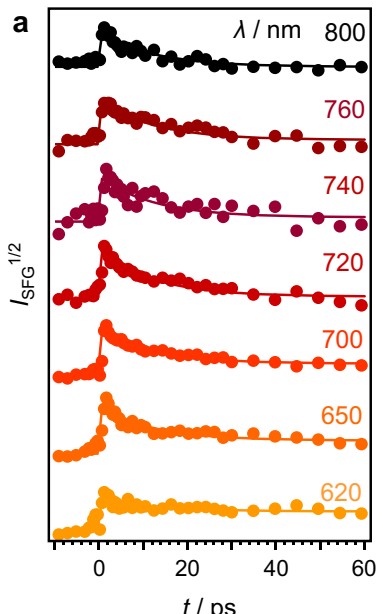

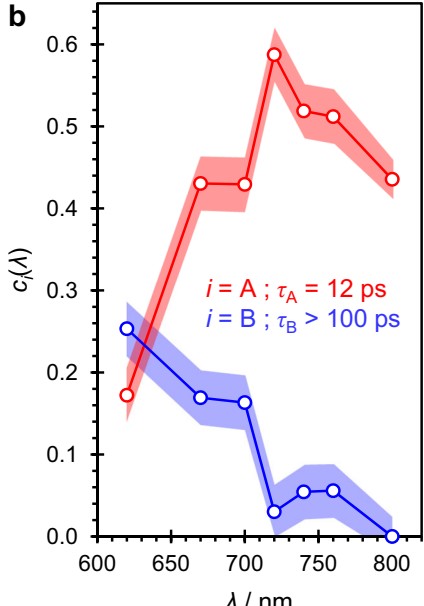

**Fig. 2 | Time- and frequency-resolved SFG signals. a** Kinetic traces for SFG signal following photooxidation of phenoxide at water/air interface at a range of wavelengths for $\omega_1$. Data points are experimentally obtained and solid lines represent a fit to a global fitting model, which in **b** yield decay-associated spectra, $c_i(\lambda)$, (as a function of $\omega_1$ wavelengths) associated with two species (A and B) decaying with lifetimes, $\tau_i$, as indicated. The shaded areas correspond to a standard deviation.

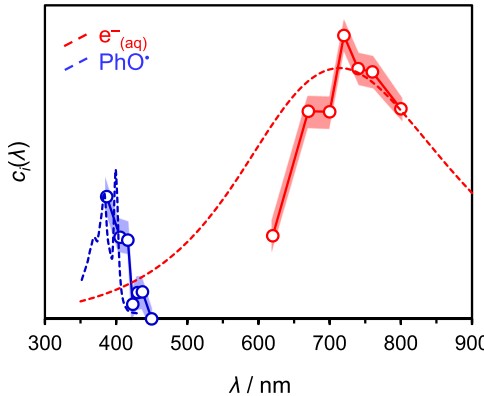

**Fig. 3 | Electronic spectra of hydrated electron and phenoxyl radical.** Data points and solid lines represent the decay-associated spectra of the hydrated electron and phenoxyl radical at the water-air interface. Dashed lines are the absorption spectra for both species in bulk environments. The shaded areas correspond to one standard deviation.

the peak positions coinciding within the experimental resolution. Hence, we conclude that species A corresponds to $e^-_{(aq/air)}$ and the spectrum measured through SFG is comparable to the absorption spectrum of $e^-_{(aq)}$. Also included in Fig. 3 is the absorption spectrum of $PhO^{\bullet}$ in aqueous solution[48], along with the spectrum of interfacial species B, where we have added the energy of $\omega_2$ to the tunable $\omega_1$. The agreement shows that $\omega_{SFG}$ is resonant with $PhO^{\bullet}$, also leading to the enhancement of the SFG signal. The absolute intensities of $c_i(\lambda)$ are not quantitatively comparable to the absorption spectra for $e^-_{(aq)}$ and $PhO^{\bullet}$, and the latter two have been scaled in Fig. 3 to aid comparison. (The maximal molar extinction coefficients are $\varepsilon_{e(aq)} = 2.3 \times 10^4\,M^{-1}\,cm^{-1}$ and $\varepsilon_{PhO^{\bullet}} = 3.0 \times 10^3\,M^{-1}\,cm^{-1}$, respectively.) Contributions from phenoxide excited states can be discounted: the $S_1$ excited state absorption peak is around 515 nm and emission around 340 nm, neither of which are resonant with $\omega_1$, $\omega_2$, or $\omega_{SFG}$; the $S_2$ excited state has an absorption or emission spectrum that is not known, but it has a sub-picosecond lifetime (leading to $e^-_{(aq)}$ and $PhO^{\bullet}$).

## Spectrum of $e^-_{(aq/air)}$

The decay-associated SFG spectrum of $e^-_{(aq/air)}$ resembles the absorption spectrum of $e^-_{(aq)}$, with the peak position being almost identical. This is expected for an electron residing at the interface but with most its electron density within the solvent, akin to water cluster anions with the highest vertical detachment energies[49]. It is also in agreement with previous conclusions from certain second-order non-linear experiments[22,42] and with theoretical predictions[50]. If the electron were to reside in an orbital that was partially hydrated, protruding out of the liquid and into the vapor phase, then the overall orbital size would be larger, with a concomitantly smaller $p \leftarrow s$ transition energy (red-shifted absorption maximum)[37,38,49]. While the peak position is similar, the spectrum of $e^-_{(aq/air)}$ appears to be narrower on the blue-edge compared to $e^-_{(aq)}$. This may be a consequence of the non-linear spectral response based on the hyperpolarizability, which is fundamentally different to the absorption spectrum (see Supplementary Note 2). Alternatively, it may arise because the blue edge of the spectrum is associated with excitation to more diffuse orbitals[13], which are likely to be perturbed at the interface and raise interesting questions about how the conduction band of water is perturbed at the water/air interface.

We also consider the effect of $PhO^{\bullet}$ that remains following photoexcitation. In the bulk, phenoxide photo-oxidation leaves $e^-_{(aq)}$ in close proximity to $PhO^{\bullet}$ and both are formed as a contact pair, $[e^-:PhO^{\bullet}]_{(aq)}$[45–47]. The absorption spectrum for the electron in such a contact pair is virtually identical to that of the free $e^-_{(aq)}$, as demonstrated by previous transient absorption spectroscopy[45–47]. The same appears to be true at the water/air interface with the presence of $PhO^{\bullet}$ showing little effect on the $e^-_{(aq/air)}$ peak position. The spectrum of the $PhO^{\bullet}$ itself appears to be red-shifted compared to the bulk solution. This is likely a result of the UV excitation wavelength used, which also accesses the second excited state of phenoxide leading to some population of $PhO^{\bullet}$ appearing in an electronically excited state that has an absorption spectrum peaking at $\lambda \approx 427\,nm$[47].

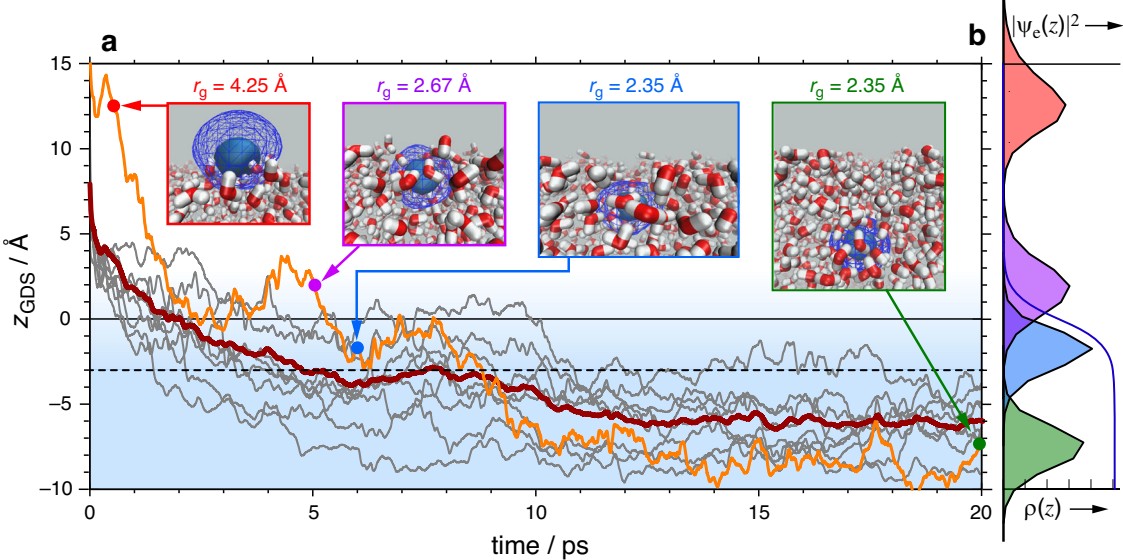

**Fig. 4 | Trajectory simulations of a hydrated electron formed at the water/air interface. a** Position of the electron's centroid ($z_{GDS}$), relative to the instantaneous Gibbs dividing surface, plotted as a function of time for each of ten trajectories (in gray and orange), with the average trajectory shown in dark red. For visual clarity, rapid fluctuations due to O–H vibrations have been smoothed using a 100 fs moving average. Trajectories are initiated from a neat interface at $t = 0$.

Representative snapshots are selected from the trajectory in orange, with opaque and mesh isocontours that encapsulate 50% and 90% of the probability $|\psi_{el}|^2$, respectively. **b** Probability distribution along the surface normal (i.e., $|\psi_{el}|^2$ integrated over lateral directions $x$ and $y$) at time points corresponding to the four snapshots. The blue curve shows the water density $\rho(z)$, where half of the bulk value defines the Gibbs dividing surface.

## Dynamics of $e^-_{(aq/air)}$

While the spectroscopy and thus the structure of the local solvation environment around the electron is very similar for $e^-_{(aq)}$ and $e^-_{(aq/air)}$, the kinetics are clearly not. From transient absorption measurements in the bulk, loss of $e^-_{(aq)}$ signal arises from geminate recombination of $[e^-:PhO^\bullet]_{(aq)}$ to reform the phenoxide anion, with a fraction also dissociating to form the free $e^-_{(aq)}$ and PhO$^\bullet$ (yield of $e^-_{(aq)} \approx 40\%$ for phenoxide excited at 257 nm)[46]. Critically, loss of $e^-_{(aq)}$ signal in the bulk is correlated with loss of PhO$^\bullet$. In contrast, the $e^-_{(aq/air)}$ signal at the interface decays with a lifetime $\tau_A = 12$ ps, which is at least an order of magnitude faster than the decay of PhO$^\bullet$. Indeed, from the minimal decay in the signal at 620 nm in Fig. 2a, we can exclude geminate recombination as a major decay mechanism, demonstrating the dramatic difference in the overall photochemistry at the interface compared to the bulk. Potential sources of loss of $e^-_{(aq/air)}$ through chemical reactions include: scavenging by $H_3O^+$, but this is in very low concentration in the present experiment; by PhO$^-$, but this seems unlikely as the dianion would not readily form; or by Na$^+$, but this resides below the surface and will not seek to form Na. Therefore, as geminate recombination is the only sub-nanosecond decay mechanism of $e^-_{(aq)}$ in the bulk, the differing dynamics observed for $e^-_{(aq/air)}$ is likely associated with a physical rather than chemical process.

At the interface the electron can diffuse into the bulk, $e^-_{(aq/air)} \rightarrow e^-_{(aq)}$. In such a scenario, the SFG signal would disappear because $e^-_{(aq)}$ would enter a centrosymmetric environment, rendering it insensitive to the second-order non-linear spectroscopic probe. The root-mean-square distance traveled for one-dimensional diffusion can be estimated as $z \approx (2Dt)^{1/2}$, where $D$ is the diffusion coefficient. Taking $D = 4.9 \times 10^{-5}$ cm$^2$ s$^{-1}$ for $e^-_{(aq)}$[51], we find that $z = 3.4$ Å for the process $e^-_{(aq/air)} \rightarrow e^-_{(aq)}$ (with $t = 12$ ps). This distance is comparable to both the size of $e^-_{(aq)}$ (radius of gyration $r_g = 2.45$ Å) and to the distance over which symmetry is broken at the water/air interface (see below and Supplementary Note 2)[52].

To support these observations and to provide deeper insight into the dynamics, we reanalyzed atomistic simulations of $e^-_{(aq/air)}$ by Coons et al.[26], which are based on the one-electron Turi-Borgis model[53]

that captures numerous physical properties of $e^-_{(aq)}$[3,4,53]. Importantly, these include the localization timescale following photochemical generation of $e^-_{(aq)}$[3], for which the model agrees with all-electron ab initio calculations[54–56], along with the experimental partial molar volume of $e^-_{(aq)}$[57], which is associated with the excluded volume occupied by $e^-_{(aq)}$. Whereas the computational expense of ab initio calculations limits simulations to typical timescales of $\lesssim 10$ ps, the one-electron model allows us to run multiple trajectories of 20–30 ps.

In these quantum/classical trajectory simulations, a liquid/vacuum interface is modeled using a periodic slab of water. (The vacuum is a good model for ambient air on the timescale of the simulations and experiments, as discussed in Supplementary Note 3). Water molecules are described in atomistic detail and the one-electron wave function, $\psi_e$, is computed on a real-space grid[26,58]. A diffuse electron is introduced at $t = 0$, where it is weakly bound to dangling O–H moieties. Figure 4a plots the position of the electron's centroid along the surface normal ($z_{GDS}$), relative to an instantaneous Gibbs dividing surface that is updated at each step and defines the interface ($z_{GDS} = 0$)[26]. Results are shown for ten different trajectories and their average, with snapshots of the electron distribution illustrated at representative points along one trajectory. The diffuse electron density (first snapshot in Fig. 4a) localizes and becomes solvated at the interface in <1 ps, consistent with sub-picosecond localization of a conduction-band electron introduced into liquid water[54–56,59]. Although these initial dynamics are not comparable to the photo-oxidation studied here, localization is nevertheless driven by formation of electron–water hydrogen bonds that are already evident within the first 0.5 ps. Subsequent dynamics reflect those associated with $e^-_{(aq/air)}$. After 1 ps, the electron's size is $r_g \approx 2.9$ Å, and after 5 ps it has settled to a roughly constant value of $r_g = 2.5 \pm 0.1$ Å. Simultaneously, $e^-_{(aq/air)}$ begins to diffuse into the bulk phase, with a centroid that hovers near $z_{GDS} \approx -3.0$ Å until $t \approx 10$ ps, which previous theoretical studies have gauged to be the lifetime of $e^-_{(aq/air)}$[26,60]. The value $z_{GDS} = -3.0$ Å (indicated by a dashed line in Fig. 4a) is significant insofar as it demarcates the boundary of the interfacial region, where the water density is 99.8% of its bulk value. Up until ~10 ps, where $z_{GDS} \approx -3.0$ Å and $r_g \approx 2.5$ Å,

much of the electron distribution remains in a non-centrosymmetric environment (third snapshot in Fig. 4a, b) and would thus be observable in the SFG experiment. Beyond -10 ps, the electron migrates further into the bulk with $z_{GDS} \lesssim -5.0$ Å and at this stage, the vast majority of the electron's probability distribution $|\psi_e|^2(t)$ resides in the centrosymmetric bulk region (fourth snapshot in Fig. 4a, b), where it is no longer be observable in a SFG experiment.

Improvements in ab initio electronic structure software have made many-electron simulations of the hydrated electron more feasible in recent years, albeit over short time scales. Remarkably, these simulations largely validate the detailed predictions of the one-electron Turi-Borgis model[3,61], confirming the veracity of the particle-in-a-cavity model. In the present work, the timescale for the $e^-_{(aq/air)} \rightarrow e^-_{(aq)}$ conversion is in excellent agreement with $\tau_A$ determined by the experiment, suggesting that the simulations have captured the overall process even though they do not contain PhO˙ nor sodium ions. The simulations additionally underscore the surface sensitivity of the SFG experiment.

### Implications

The conclusion that $e^-_{(aq/air)}$ is fully solvated, with only a fractional electron density exposed to the vapor phase, brings into question the interpretation of studies suggesting a partially hydrated electron with most of its density protruding into the vapor phase[19,21]. Whether this is so has important consequences from a chemical reactivity perspective. A more diffuse density extending into the vapor phase would have different energetics and would lie in an energy range commensurate with electron attachment to molecules including DNA, with the possibility to induce strand cleavage[2,19]. It is also interesting to compare our results to photoelectron spectroscopy of $e^-_{(aq/air)}$ on liquid microjets, where the photoelectron signal appears to support our conclusion that $e^-_{(aq/air)}$ is solvated below the interface, but where the electron is observed to reside at the interface for longer than observed here[20,27–32]. While the composition between air versus vacuum is of little significance on the timescales of the current experiment and simulations (see Supplementary Note 3), the nature of the surface and the probe-depth of the spectroscopic method are important. In the current ambient-condition experiment, the Gibbs dividing surface is well-defined, whilst in the case of a liquid microjet, evaporation from the surface is likely to distort this, as evidenced by non-thermal distributions of evaporated molecules[62]. Additionally, the probe depth for the SFG experiment is on the order of 3 Å, as shown in the current experiment and governed by the asymmetry of the water environment. In photoelectron spectroscopy, the probe depth is dictated by the effective attenuation length of an electron in liquid water, which depends on the energy of the outgoing electron and is at best on the order of a few nm for energies between 10 and 100 eV, although precise values are still debated[63]. In any case, such experiments are not sensitive to the $e^-_{(aq/air)} \rightarrow e^-_{(aq)}$ dynamics.

In contrast to the fast internalization dynamics of $e^-_{(aq/air)}$, PhO˙ remains at the interface for much longer times (>100 ps), suggesting that either the contact pair dissociates very rapidly or else is never formed in the first place. The persistence of PhO˙ at the water/air interface suggests that it may be reactive with other chemical species in the vapor phase or at the interface. Indeed, a proposed mechanism for chemical rate enhancements observed at the surface of aqueous microdroplets includes the removal of $e^-_{(aq/air)}$, via diffusion to the bulk, as one step[10]. In this model, OH⁻ is ionized by strong interfacial electric fields, leaving reactive OH˙ at the interface once $e^-_{(aq/air)}$ has diffused away. While making no comment on the validity of this proposed mechanism, the diffusive $e^-_{(aq/air)} \rightarrow e^-_{(aq)}$ step is consistent with our observations. Viewed more generally, the interface acts as an effective separator for the two reactants, leaving both radicals in distinct environments where they can then potentially undergo further reactions.

## Discussion

The optical spectrum of $e^-_{(aq/air)}$ is similar to that of $e^-_{(aq)}$, demonstrating that most of the electron density resides within the aqueous phase rather than the vapor phase as suggested in certain previous studies. The implication is that the electron by itself is no more reactive at the interface than in the bulk. While spectroscopically similar, the dynamics of $e^-_{(aq/air)}$ differ, as it diffuses rapidly into the bulk leaving behind its molecular parent, which in the present study is the phenoxyl radical. The latter remains at the surface where it could participate in reactivity with vapor-phase species, with potential implications for reactivity in microdroplets and in atmospheric chemistry. More generally, the water/air interface also serves as a general model for a hydrophobic interface, suggesting that the ultrafast radical separation dynamics may be common at many aqueous interfaces. From an experimental viewpoint, the spectral and mechanistic insight gained here were only made possible by directly probing all products at the water/air interface, demonstrating the potential of time-resolved electronic SFG as a method for probing interfacial dynamics, in much the same way that transient absorption spectroscopy has become a workhorse technique to probe bulk dynamics.

## Methods
### Experimental

The time-resolved sum-frequency generation spectroscopy arrangement has been detailed in Ref. 64. The output of an Yb:KWG laser (Light Conversion, Carbide 5, producing 230 fs pulses at 1026 nm with 83 μJ pulse⁻¹ energy at 12 kHz) was split into three parts. One part was used to generate pump pulses, $\omega_{pump}$, at 257 nm (1.3 μJ pulse⁻¹) by frequency quadrupling in two successive BBO crystals. The pump was chopped at 6 kHz to enable active pump-on/pump-off subtraction. A second part was used for light field $\omega_2$ ($\lambda = 1026$ nm). A third part was used to pump an optical parametric amplifier (Light Conversion, Orpheus) producing tuneable light $\omega_1$ ($620 \leq \lambda \leq 800$ nm). Light fields $\omega_1$ and $\omega_2$ were collinearly combined and focussed onto the liquid surface ($f = 20$ cm at an angle of incidence of 73°). Fields $\omega_1$ and $\omega_2$ were temporally overlapped and delayed relative to $\omega_{pump}$ using a motorized delay stage. The resultant field, $\omega_{SFG}$, was separated from $\omega_1$ and $\omega_2$ using a Pelin-Broca prism and sent to an optical Kerr gate, where fluorescence from the sample induced by $\omega_{pump}$ was suppressed. The $\omega_{SFG}$ was subsequently collected using a photomultiplier tube (Hamamatsu H7732-10), the output of which was electronically gated and discriminated (Advanced Research Instruments F-100TD), and pulses counted on two separate counters for pump-on and pump-off measurements. Count rates were <10⁻² photons/shot and transients are typically collected over 10⁶ laser shots/delay. Polarizations of $\omega_1$, $\omega_2$, and $\omega_{SFG}$ were set to PPP. The pump was also P-polarized.

Specific care was taken to ensure measurements at differing wavelengths were comparable. The resonant signal contribution was normalized to the nonresonant background signal present in each of the pump-off traces such that the only difference between pump-on to the pump-off channels was the presence of the excited species at the interface, which was affected by pump-probe overlap, sample concentration, and pump power. The sample concentration was kept constant between measurements, with an approximate maximum error of 5%. The pump energy also varied no more than 5% within, and between, datasets. The main source of errors are the spatial overlap between the pump and probe pulses and any changes in the divergence of the tunable $\omega_1$ field, leading to changes in the focus at the water/air interface. To minimize this, the overlaps and spot-sizes were independently monitored using a 10-fold digital microscope.

The sample (-75 ml of 150 mM phenol (Sigma Aldrich) in water (18 MΩ cm, Millipore), made to pH 13 using NaOH (Sigma Aldrich) to promote deprotonation) was contained in a rotating (0.5 rev s⁻¹) petri-dish and the surface height was kept to ± 14 μm using a home-built

liquid-height monitor. The surface coverage of phenoxide was ~7% (see Supplementary Note 4).

## Computational

Quantum/classical molecular dynamics simulations were performed as detailed in previous work[26], using the electron–water pseudopotential developed by Turi and Borgis[53] and a periodic slab geometry containing 200 SPC water molecules at normal liquid density. The neat liquid slab was equilibrated at 300 K, following which an electron is introduced to define $t = 0$. Atomistic dynamics are then propagated (using Ewald summation in a 18.1722 Å × 18.1722 Å × 54.5166 Å unit cell), using a Nosé-Hoover thermostat and a 1 fs time step. The one-electron Schrödinger equation is solved on a real-space grid at every step, to obtain adiabatic forces for molecular dynamics. The grid points span the liquid slab and extend well into the vacuum, with a spacing $\Delta x = \Delta y = 0.947$ Å and $\Delta z = 0.971$ Å. These simulation parameters are well-tested for obtaining converged dynamics[26,58].

## Data availability

The experimental and trajectory data generated in this study have been deposited in the Zenodo database under the accession code https://doi.org/10.5281/zenodo.8005779.

## Code availability

The source code used in the trajectory simulations is available in the aforementioned Zenodo database.

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

## Acknowledgements

We thank Faith Prichard for her support with parts of the experimental work. This work was supported by the Engineering and Physical Sciences Research Council [grant number EP/R513039/1] (C.J.C.J.) and the U.S. National Science Foundation grant CHE-1955282 (J.M.H.).

## Author contributions

J.R.R.V. conceived the overall project. C.J.C.J. and J.R.R.V. conceived the experimental methodology and C.J.C.J. performed the experiments and data analysis. J.M.H. conceived the computational methodology, M.P.C. performed the calculations and both M.P.C. and J.M.H. performed the data analysis. J.R.R.V. wrote the manuscript with input from all authors.

## Competing interests

The authors declare no competing interests.
