## [Peer Review File · Nature Communications]

Reviewers' Comments:

Reviewer #1:

Remarks to the Author:

This is very nice work, and certainly ought to be published.

I only found one minor flaw. The spectrum of aqueous surface phenoxyl radical deduced in this work is compared for some reason with matrix isolated phenoxyl. The spectrum in bulk water has been known for a long time, and it is far more appropriate to compare directly with the bulk water spectrum. I can only surmise that the authors did not know where to look. The spectrum is included in this paper:

Spectral characteristics of monosubstituted phenoxyl radicals

Wojnarovits, L; Kovacs, A and Foldiak, G

Radiation Physics and Chemistry 50 (4) , pp.377-379

There is a digital version in an older cited paper:

SCHULER, RH and BUZZARD, GK

1976 |

INTERNATIONAL JOURNAL FOR RADIATION PHYSICS AND CHEMISTRY 8 (5) , pp.563-574

I am sure there must be other sources as well.

Publish after fixing this flaw.

Reviewer #2:

Remarks to the Author:

This paper presents an investigation of solvated electrons at the water/air interface. Using time-resolved electronic sum-frequency generation (SFG) the authors found evidence for the spectra and the lifetimes of the byproducts of the photo-excitation of phenoxide anions at the interface, namely, solvated electrons and phenoxyl radicals. The authors argue that the decay-associated SFG spectrum of solvated electrons at the interface resembles the absorption spectrum of a solvated electron in bulk water, indicating that most of its electron density resides in the liquid phase. Additionally, the study reports a solvated electron lifetime at the interface of 12 ps, and a phenoxyl radical lifetime greater than 100 ps. By ruling out any other possible mechanism for the disappearance of the solvated electron signal, the authors attribute the 12-ps lifetime to the escape of the electron from the interface into the bulk. To support this claim, the authors performed molecular dynamics simulations of the solvated electron at the water/vacuum interface, finding that the solvated electron leaves the interface within around 10 ps.

The paper addresses one of the least understood aspects of the solvated electron, namely, its behavior at the water/air interface. By providing the most direct experimental evidence so far of the solvated electron's behavior at the interface, the study demonstrates that the solvated electron is repelled by the water/air interface and, thus, gives insight into possible applications in atmospheric chemistry. However, the relevance of studying the solvated electron at an interface is barely mentioned in the introduction, which weakens the presentation of the motivation for the present study. Aside from this minor remark, the paper is well written and the results are clearly presented. SFG is well established for the characterization of interfaces and the simulations have demonstrated their capability of reproducing the experimental results. We find the paper to be of high quality and will likely recommend it for publication in Nature Communications after the authors have addressed the following remarks. (This report was prepared by an established referee together with an early-career researcher.)

(1) In the manuscript, geminate recombination is ruled out as a possible mechanism for the decay of the solvated electron signal adducing that the signal at 620 nm in Fig. 2(a) barely decays. Although an explanation on why phenolate anions cannot contribute to the signal is provided, it only appears in the supporting material. In our opinion, the authors should expand on this point in the main body of the manuscript, as it is crucial for the interpretation of the results. Also, in the

explanation provided in the supporting material (p. 3), the authors reference an article that does not mention phenolate at all, which is confusing.

(2) The role of the air in the water/air interface is not clear. Would a water/vacuum interface not have produced the same results? The authors emphasize the difference between the water/air interface and the water/vacuum interface in the introduction. On p. 9 of the manuscript, the authors write that they performed atomistic simulations of e_{water/air} as in Coons et al., J. Am. Chem. Soc. 2016, 138 (34). However, in that reference it is clearly stated that the simulations were done assuming a water/vacuum interface.

(3) On p. 4 there seems to be a typo: "As omega_2 was scanned across ..." should be replaced with "As omega_1 was scanned across ..."

(4) The authors should provide a short description of the fitting procedure employed.

(5) The computational methodology employed is poorly explained in the main body of the manuscript. Given the short length of this section in the supporting material, it should be included in the main body.

(6) With the advent of new all-electron methods to study electron solvation in water, it is remarkable that a one-electron potential yields such apparently good results. Therefore, the authors should comment on the limitations of using a one-electron potential to study electron solvation, as there have been instances where such potentials have failed to reproduce experimental results.

(7) There is an author missing in the author list of the supporting material (Marc P. Coons).

Reviewer #3:

Remarks to the Author:

Reviewer #4:

Remarks to the Author:

The manuscript by Jordan et al. is aimed at determining the absorption spectrum and time-dynamics of the interfacial aqueous electron. There is a body of literature related to the same phenomenon in clusters but not for systems in which the electron can possess either extended delocalization or can diffuse over large distances as is possible by studying the liquid water interface. Interest in the interfacial aqueous electron comes from many quarters: from a fundamental point of view, the question is whether it occupies a cavity akin to its bulk counterpart or whether the electron density extends into the vapour phase, and for how long it resides at the interface; the latter is a controversial quantity which requires both knowledge of the reactivity of the interfacial species, as well as the time for diffusion. These quantities have historically been difficult to conclusively, with a single experiment, disentangle. But there are several more applied reasons that the interfacial reactivity of hydrated electrons can be important – from recently proposed reduction reactions at the surface of microdroplets, radiation chemistry, to the ocean interface and even surface plasmon driven electron chemistry. It is for this reason that Nature Communications is an appropriate venue for publishing this exciting result as we expect it to be of interest to scientists in several disciplines. Overall, the work is strong and the paper should be published after the authors consider the following comments and suggested revisions.

The current study builds on an earlier rapid communication from Verlet (ref. 35), to now provide insight on both the spectrum of the hydrated electron at the interface, as well as its reaction dynamics with respect to ejection, recombination and migration into the bulk. Importantly this

manuscript adds strong support from theory from Herbert as to the behaviour of an electron placed onto the water network. In the experiment, a molecular anion, phenolate, with a low photodetachment energy is chosen to launch the electron in the surface layer, because of its claimed surface activity (see below).

In motivating their work, the authors mention other techniques typically used to differentiate different electron-binding motifs such as photoelectron spectroscopy, however the authors have eschewed liquid jet PE spectroscopy using the argument that one is probing a liquid/vacuum interface (liquid microjet PES requires in-vacuo microjets to limit air scattering) rather than the liquid-air interface. One wonders if this contrast is rather overplayed, as the atmospheric number density of air molecules means that over the surface area relevant to a single hydrated electron, there is likely to be 1 or less vapor molecule. (Putative differences between liquid-vacuum versus air-water interfaces would be interesting for the authors to expand on - or provide references to supporting work suggesting the important differences in water structure and dynamics at these interfaces).

A perhaps more important reason to apply an alternative experimental approach is the shorter probing depth being sampled. Electronic SFG, used here, is uniquely sensitive to environments for which centro-symmetry is broken; for a small solute like the electron or phenoxide species, that is a few water layers closest to the interface (shown nicely in Figure 4). In contrast, signals from bulk-residing species do not appear.

The biggest overall concern that it would be good to see the authors fully address: that is whether the E-SFG response should actually resemble the linear electronic absorption spectrum. This is currently just hinted on page 7 ("this may be a consequence of the non-linear spectral response based on the hyperpolarizability"), but the E-SFG response should be connected to a different matrix element than simply the dipole moment transition element. In low symmetry cases, this may not make a great deal of difference but for an "atomic" transition largely built from s to p

character, this will be more subtle. Perhaps a section in the SI that addresses what would be expected for the energy dependent $\chi(2)$ response?

Comments on specific point:

- In the experiment ω_{pump} is chosen to be at 257 nm to facilitate the $S_1 \rightarrow S_0$ transition of phenolate. We realize that establishing the mechanism as autodetachment is the subject of a different paper from Verlet, however, it might be helpful to the reader to state that an electron is ejected as part of this photoexcitation which subsequently leads to the formation of a fully solvated electron.

- "Therefore, two driving fields with frequencies ω_1 and ω_2 will combine to generate the SFG field with frequency $\omega_{\text{SFG}} = \omega_1 + \omega_2$, exclusively from the interface. The field ω_{SFG} can be enhanced when any of the three fields" – up to this point, only two fields $\omega_1 + \omega_2$ have been mentioned.

- Another clarification for readers who are not specialists: "Both ω_1 and ω_2 were delayed together with respect to ω_{pump} ..."

- Literature relating to the surface-activity of phenolate should be cited; phenol is surface active in water and phenolate is likely to be surface active too, however supporting literature for this should be cited.

- "Specific consideration was given to experimental parameters between measurements at different ω_1 to ensure that the relative signals measured were comparable." – this is rather too vague. Which experimental parameters are suggested here?

- The following sentence needs to be clarified: "In the limit of weak non-resonant signal..." What is non-resonant? The surface water signal?

My first reading of this sentence prompted the following question: If non-resonant SFG signal intensity depends quadratically on the concentration of absorbers, how much enhancement is

expected of the resonant transition i.e. what is the level of signal expected above a non-resonant background if a resonant transition is present? But on re-reading, perhaps the authors were trying to say how the signal scaled for the resonant transition. Please clarify what is the non-resonant part.

- "As ω_2 was scanned across a range of the absorption spectrum of $e^-(aq)$ " – this seems to contradict the text that ω_2 was at a fixed wavelength of 1026 nm. What were the experimental reasons for why the wavelength ω_2 was not chosen to be at the absorption maximum (720 nm) of the bulk hydrated electron at room temperature?

- Why is the Argon matrix spectrum of PhO being primarily cited and reproduced in Figure 3 – there are perfectly good absorption spectra (in papers already referenced by the authors) for the phenoxyl radical spectra in water that are much more relevant to the current study. Figure 3 would be more complete if the bulk water versus interfacial water

spectra were compared for both partners.

- "as ω_1 blue-shifts" blue shift is usually used to refer to a response of a system rather than the action of sweeping a wavelength

-

- "As geminate recombination is the only sub-nanosecond decay mechanism of $e^-(aq)$ in the bulk, the differing dynamics observed for $e^-(aq/air)$ is likely not associated with a chemical process, but with a physical process" This deduction is not so clear to the reviewer; are the authors able to rule out that a new chemical channel might open up at the interface, perhaps because of the enhanced reactivity of the interfacial electron or from an enhanced interfacial concentration of hydronium ions, with which the electrons can react?

- Other work using vibrational sum frequency generation (Imamura et al. Phys. Chem. C 2014, 118, 50, 29017–29027) has concluded that hydroxide ions are not surface active. This relates to this work indirectly since decomposition of the solvated electron leads to hydroxide formation, which would appear at the interface as a transient if the reactivity of the interfacial electron is quicker than its timescale for diffusion. Can an electronic SFG signature of -OH be ruled out of these data?

- The broadness of the absorption spectrum (pertaining to the s-p transition) of the bulk aqueous electron is attributed to the lifting of the degeneracy of the three p-orbitals caused by a dynamical, non-perfectly spherical cavity. For an electron near an interface, where the cavity, if it exists at all as a conventional cavity, is likely to deviate substantially from spherical, wouldn't the energy spacings between the three directional p-orbitals involved in the s-p transition diverge even further, leading to a broader spectrum for the interfacial aqueous electron than the bulk? The experiment sees the reverse. Can the authors perhaps further speculate on the bandwidth for the species that they have attributed to the interfacial electron in Figure 3?

- For the blue side, the authors provide the following "or it may arise because the blue edge of the spectrum is associated with excitation to more diffuse orbitals". For the bulk aqueous electron, the asymmetric absorption tail into the blue is associated with transitions to the conduction band of water contributing at the shortest wavelengths. For this data, it would seem not only is the peak narrowed on the blue side, but that there is no blue-most 'tail' either; the absorption simply plummets rapidly at 600 nm. This is interesting as it could indicate that the conduction band is fundamentally altered in an interfacial experiment.

Reviewer #5:

Remarks to the Author:

I co-reviewed this manuscript with one of the reviewers who provided the listed reports. This is part of the Nature Communications initiative to facilitate training in peer review and to provide

appropriate recognition for Early Career Researchers who co-review manuscripts.

Dear Reviewers,

We are grateful for your positive view of our work, your careful reading, and your useful suggestions, which we have taken into account in a revised manuscript (one with changes tracked and one with all changes accepted). Below, we address all of your comments (blue text).

Note that in the revised manuscript, we have made some very minor edits to improve readability. We have not tracked those changes to make it easier for you to correlate comments with changes made in the text (or Supplementary Information).

REVIEWER COMMENTS

Reviewer #1 (Remarks to the Author):

This is very nice work, and certainly ought to be published.

I only found one minor flaw. The spectrum of aqueous surface phenoxyl radical deduced in this work is compared for some reason with matrix isolated phenoxyl. The spectrum in bulk water has been known for a long time, and it is far more appropriate to compare directly with the bulk water spectrum. I can only surmise that the authors did not know where to look. The spectrum is included in this paper:

Spectral characteristics of monosubstituted phenoxyl radicals
Wojnarovits, L; Kovacs, A and Foldiak, G
Radiation Physics and Chemistry 50 (4) , pp.377-379

There is a digital version in an older cited paper:

SCHULER, RH and BUZZARD, GK
1976 |

INTERNATIONAL JOURNAL FOR RADIATION PHYSICS AND CHEMISTRY 8 (5) , pp.563-574

I am sure there must be other sources as well.

Publish after fixing this flaw.

We thank the reviewer for the suggestions. We are happy to make the change although do note that the phenol radical is only partially solvated so the “true” spectrum may lie somewhere in between. Nevertheless, we have updated the figure and the relevant references.

Reviewer #2 (Remarks to the Author):

This paper presents an investigation of solvated electrons at the water/air interface. Using time-resolved electronic sum-frequency generation (SFG) the authors found evidence for the spectra and the lifetimes of the byproducts of the photo-excitation of phenoxide anions at the interface, namely, solvated electrons and phenoxyl radicals. The authors argue that the decay-associated SFG spectrum of solvated electrons at the interface resembles the absorption spectrum of a solvated electron in bulk water, indicating that most of its electron density resides in the liquid phase. Additionally, the study reports a solvated electron lifetime at the interface of 12 ps, and a phenoxyl radical lifetime greater than 100 ps. By ruling out any other possible mechanism for the disappearance of the solvated electron signal, the authors attribute the 12-ps lifetime to the escape of the electron from the interface into the bulk. To support this claim, the authors performed molecular dynamics simulations of the solvated electron at the water/vacuum interface, finding that the solvated electron leaves the interface within around 10 ps.

The paper addresses one of the least understood aspects of the solvated electron, namely, its behavior at the water/air interface. By providing the most direct experimental evidence so far of the solvated electron's behavior at the interface, the study demonstrates that the solvated electron is repelled by the water/air interface and, thus, gives insight into possible applications in atmospheric chemistry. *However, the relevance of studying the solvated electron at an interface is barely mentioned in the introduction, which weakens the presentation of the motivation for the present study.*

This is a fair comment and we had initially focussed the paper on the key physical questions. We have briefly commented on the broader significance of interfacial electrons now in the introduction.

Aside from this minor remark, the paper is well written and the results are clearly presented. SFG is well established for the characterization of interfaces and the simulations have demonstrated their capability of reproducing the experimental results. We find the paper to be of high quality and will likely recommend it for publication in Nature Communications after the authors have addressed the following remarks. (This report was prepared by an established referee together with an early-career researcher.)

(1) In the manuscript, geminate recombination is ruled out as a possible mechanism for the decay of the solvated electron signal adducing that the signal at 620 nm in Fig. 2(a) barely decays. Although an explanation on why phenolate anions cannot contribute to the signal is provided, it only appears in the supporting material. In our opinion, the authors should expand on this point in the main body of the manuscript, as it is crucial for the interpretation of the results. Also, in the explanation provided in the supporting material (p. 3), the authors reference an article that does not mention phenolate at all, which is confusing.

We have moved a shortened version into the main text. The more detailed discussion is retained in the supporting information for completeness. As for the reference that does not relate to phenolate, the reviewer is right and we accidentally referenced the incorrect paper by the Bradforth group. We have corrected this.

(2) The role of the air in the water/air interface is not clear. Would a water/vacuum interface not have produced the same results? The authors emphasize the difference between the water/air interface and the water/vacuum interface in the introduction. On p. 9 of the manuscript, the authors write that they performed atomistic simulations of e_{water/air} as in Coons et al., J. Am. Chem. Soc. 2016, 138 (34). However, in that reference it is clearly stated that the simulations were done assuming a water/vacuum interface.

The water/air interface is effectively the same as a water/vacuum interface (except for the evaporation problem). This is because the collision frequency for a gas-phase molecule with the surface is about once every ns, the timescale of which exceeds that considered in the current work. We have added a brief comment to consider this (and the comments from Reviewer #4/5) and have added a discussion to justify this in the Supplementary Information.

(3) On p. 4 there seems to be a typo: "As omega_2 was scanned across ..." should be replaced with "As omega_1 was scanned across ..."

Yes, great spot – thank you (see also comment by Reviewer #4/5)

(4) The authors should provide a short description of the fitting procedure employed.

We have added a discussion of this in the Supporting Information.

(5) The computational methodology employed is poorly explained in the main body of the manuscript. Given the short length of this section in the supporting material, it should be included in the main body.

While we appreciate the desire to understand the details of the calculations, we feel that a description of the methodology in the main text would break up the flow of the paper too much. The

key point really is that the results agree very closely with the experiment. The methodology has been extensively documented in previous publications, e.g., Refs. 25 and 57. Core aspects of the simulation are provided in the main text, and a few more details have been added in the revision. However, we leave the detailed explanation of grid parameters, time steps, and boundary conditions in the Methods section, together with the experimental details.

(6) With the advent of new all-electron methods to study electron solvation in water, it is remarkable that a one-electron potential yields such apparently good results. Therefore, the authors should comment on the limitations of using a one-electron potential to study electron solvation, as there have been instances where such potentials have failed to reproduce experimental results.

It's true that all-electron DFT simulations (and even MP2, albeit at extreme cost) have become more feasible in recent years, largely thanks to improved implementations of periodic hybrid DFT in the CP2K code. From our point of view, the results from those simulations serve to confirm the veracity of carefully constructed one-electron pseudopotential models, such as the Turi-Borgis model that is used here. Despite some high-profile papers creating a controversy in the mid-2010s, the fact remains that most experimental observables are reproduced quite well by these models, as detailed by one of us in a 2019 review article (Ref. 3). A brief discussion of this point and some additional references have been added in the discussion of the trajectory simulations.

(7) There is an author missing in the author list of the supporting material (Marc P. Coons).

That's an oversight on our part – it has now been fixed.

Reviewer #3 (Remarks to the Author):

Great initiative – thank you for your contribution.

Reviewer #4 (Remarks to the Author):

The manuscript by Jordan et al. is aimed at determining the absorption spectrum and time-dynamics of the interfacial aqueous electron. There is a body of literature related to the same phenomenon in clusters but not for systems in which the electron can possess either extended delocalization or can diffuse over large distances as is possible by studying the liquid water interface. Interest in the interfacial aqueous electron comes from many quarters: from a fundamental point of view, the question is whether it occupies a cavity akin to its bulk counterpart or whether the electron density extends into the vapour phase, and for how long it resides at the interface; the latter is a controversial quantity which requires both knowledge of the reactivity of the interfacial species, as well as the time for diffusion. These quantities have historically been difficult to conclusively, with a single experiment, disentangle. But there are several more applied reasons that the interfacial reactivity of hydrated electrons can be important – from recently proposed reduction reactions at the surface of microdroplets, radiation chemistry, to the ocean interface and even surface plasmon driven electron chemistry. It is for this reason that *Nature Communications* is an appropriate venue for publishing this exciting result as we expect it to be of interest to scientists in several disciplines. Overall, the work is strong and the paper should be published after the authors consider the following comments and suggested revisions.

The current study builds on an earlier rapid communication from Verlet (ref. 35), to now provide insight on both the spectrum of the hydrated electron at the interface, as well as its reaction dynamics with respect to ejection, recombination and migration into the bulk. Importantly this manuscript adds strong support from theory from Herbert as to the behaviour of an electron placed onto the water network. In the experiment, a molecular anion, phenolate, with a low photodetachment energy is chosen to launch the electron in the surface layer, because of its claimed surface activity (see below).

In motivating their work, the authors mention other techniques typically used to differentiate different electron-binding motifs such as photoelectron spectroscopy, however the authors have eschewed liquid jet PE spectroscopy using the argument that one is probing a liquid/vacuum interface (liquid microjet PES requires in-vacuo microjets to limit air scattering) rather than the liquid-air interface. One wonders if this contrast is rather overplayed, as the atmospheric number density of air molecules means that over the surface area relevant to a single hydrated electron, there is likely to be 1 or less vapor molecule. (Putative differences between liquid-vacuum versus air-water interfaces would be interesting for the authors to expand on - or provide references to supporting work suggesting the important differences in water structure and dynamics at these interfaces).

This is an important point. The main difference is not between the “composition” of air and vacuum as we agree there will be little difference on the timescales of the simulations or experiments (see response to Reviewer #2/3 and addition to Supplementary Information). However, there is a significant difference in regards of evaporation, which will mean that the interface in liquid microjets is *not* the same as the ambient water/air interface. The extent of this difference remains somewhat uncertain (see for example J. Chem. Phys. 150, 044201 (2019)). We have added a discussion considering the differences between the photoelectron spectroscopy and our SFG experiments.

A perhaps more important reason to apply an alternative experimental approach is the shorter probing depth being sampled. Electronic SFG, used here, is uniquely sensitive to environments for which centro-symmetry is broken; for a small solute like the electron or phenoxide species, that is a few water layers closest to the interface (shown nicely in Figure 4). In contrast, signals from bulk-residing species do not appear.

We have addressed this in the above point through the corresponding addition in the manuscript and supplementary information.

The biggest overall concern that it would be good to see the authors fully address: that is whether the E-SFG response should actually resemble the linear electronic absorption spectrum. This is currently just hinted on page 7 (“this may be a consequence of the non-linear spectral response based on the hyperpolarizability”), but the E-SFG response should be connected to a different matrix element than simply the dipole moment transition element. In low symmetry cases, this may not make a great deal of difference but for an “atomic” transition largely built from s to p character, this will be more subtle. Perhaps a section in the SI that addresses what would be expected for the energy dependent $\chi^{(2)}$ response?

This is a good suggestion and, in the relevant place in the manuscript, we now refer the reader to a discussion on this point in the supplementary information. While the methods are different by nature, there are very good arguments that – on the whole – the absorption spectrum should follow the SFG spectrum. We present these arguments as well as a few examples from the literature where *electronic* SFG spectra have been compared to corresponding UV-vis spectra.

Comments on specific point:

- In the experiment ω_{pump} is chosen to be at 257 nm to facilitate the S1 \leftarrow S0 transition of phenolate. We realize that establishing the mechanism as autodetachment is the subject of a different paper from Verlet, however, it might be helpful to the reader to state that an electron is ejected as part of this photoexcitation which subsequently leads to the formation of a fully solvated electron.

We have now included this.

- “Therefore, two driving fields with frequencies ω_1 and ω_2 will combine to generate the SFG field with frequency $\omega_{SFG} = \omega_1 + \omega_2$, exclusively from the interface. The field ω_{SFG} can be enhanced when any of the **three** fields” – up to this point, only two fields $\omega_1 + \omega_2$ have been mentioned.

We have now included this.

- Another clarification for readers who are not specialists: “Both ω_1 and ω_2 were delayed **together** with respect to ω_{pump} ...”

We have now included this.

- Literature relating to the surface-activity of phenolate should be cited; phenol is surface active in water and phenolate is likely to be surface active too, however supporting literature for this should be cited.

We have now included this and have included our surface tension measurements in the Supplementary Information which agrees with previous non-linear spectroscopic work.

- “Specific consideration was given to experimental parameters between measurements at different ω_1 to ensure that the relative signals measured were comparable.” – this is rather too vague. Which experimental parameters are suggested here?

We reference the reader to the Supplementary Information which describes this in more detail: *Specific care was taken to ensure measurements at differing wavelengths are comparable. The resonant signal contribution was normalised to the nonresonant background signal present in each of the pump-off traces such that the only difference between pump-on to the pump-off channels was the presence of the excited species at the interface, which was affected by pump-probe overlap, sample concentration and pump power. The sample concentration was kept constant between measurements, with an approximate maximum error of 5%. The pump energy also varied no more than 5% within, and between, datasets. The main source of errors are the spatial overlap between the pump and probe pulses and any changes in divergence of the tunable ω_1 field, leading to changes in the focus at the water/air interface. To minimise this, the overlaps and spot-sizes were independently monitored using a 10-fold digital microscope.*

This section has now been moved into the Methods section so is easier to access for the reader.

- The following sentence needs to be clarified: “In the limit of weak non-resonant signal...” What is non-resonant? The surface water signal?
My first reading of this sentence prompted the following question: If non-resonant SFG signal intensity depends quadratically on the concentration of absorbers, how much enhancement is expected of the resonant transition i.e. what is the level of signal expected above a non-resonant background if a resonant transition is present? But on re-reading, perhaps the authors were trying to say how the signal scaled for the *resonant* transition. Please clarify what is the non-resonant part.

Yes, the re-reading is correct. The core issue in SFG arises if the non-resonant and resonant signals are of similar magnitude because the phase between the two is not known. In the case where the resonant contribution is larger than the non-resonant case (which is common and the case for the water surface where the non-resonant response is very weak) will the response be purely quadratic with concentration. We have hopefully clarified this now.

- “As ω_2 was scanned across a range of the absorption spectrum of $e^-(aq)$ ” – this seems to contradict the text that ω_2 was at a fixed wavelength of 1026 nm. What were the experimental reasons for why the wavelength ω_2 was not chosen to be at the absorption maximum (720 nm) of the bulk hydrated electron at room temperature?

This is an error on our part and the subscript should be “1” (see also comment (3) by Reviewer #2/3). ω_1 was scanned across the absorption maximum (720 nm) of the bulk hydrated electron.

- Why is the Argon matrix spectrum of PhO being primarily cited and reproduced in Figure 3 – there are perfectly good absorption spectra (in papers already referenced by the authors) for the phenoxyl radical spectra *in water* that are much more relevant to the current study. Figure 3 would be more complete if the bulk water versus interfacial water spectra were compared for both partners.

We have now replaced this with the spectrum in water (see also comment from Reviewer #1). We do note that the phenol radical is only partially solvated so the “true” spectrum may lie somewhere in between.

- “as ω_1 blue-shifts” blue shift is usually used to refer to a response of a system rather than the action of sweeping a wavelength.

We have changed the wording here.

- “As geminate recombination is the only sub-nanosecond decay mechanism of $e^-(aq)$ in the bulk, the differing dynamics observed for $e^-(aq/air)$ is likely not associated with a chemical process, but with a physical process” This deduction is not so clear to the reviewer; are the authors able to rule out that a new *chemical* channel might open up at the interface, perhaps because of the enhanced reactivity of the interfacial electron or from an enhanced interfacial concentration of hydronium ions, with which the electrons can react?

We have added a sentence to hopefully clarify by considering all other potential reactions. The concentration of H_3O^+ is very small given that the pH is 13 and so will not be important. It's not phenolate because that repels the electron and will not form a dianion. Finally, Na^+ lies below the interface, but Na^+ does not want an electron as it is well-known that Na spontaneously ionises in water.

- Other work using vibrational sum frequency generation (Imamura et al. Phys. Chem. C 2014, 118, 50, 29017–29027) has concluded that hydroxide ions are not surface active. This relates to this work indirectly since decomposition of the solvated electron leads to hydroxide formation, which *would* appear at the interface as a transient if the reactivity of the interfacial electron is quicker than its timescale for diffusion. Can an electronic SFG signature of $\cdot OH$ be ruled out of these data?

An interesting point, but we would not be resonant with any transitions in the hydroxide ion. The first absorption band is the CTTS and this lies around 6 eV – well beyond the spectral range used in the current experiment. So, it would not seem likely that OH^- contributes to the signal.

- The broadness of the absorption spectrum (pertaining to the s-p transition) of the *bulk* aqueous electron is attributed to the lifting of the degeneracy of the three p-orbitals caused by a dynamical, non-perfectly spherical cavity. For an electron near an interface, where the cavity, if it exists at all as a conventional cavity, is likely to deviate substantially from spherical, wouldn't the energy spacings between the three directional p-orbitals involved in the s-p transition diverge *even further*, leading to a broader spectrum for the interfacial aqueous electron than the bulk? The experiment sees the reverse. Can the authors perhaps further speculate on the bandwidth for the species that they have attributed to the interfacial electron in Figure 3?

It is an interesting point and one we had thought about for quite some time. However, one of the core conclusions of this work is that the cavity at the interface is in fact very similar to that in the bulk. The evidence for this from both experiment and theory is: (i) the electron resides below Gibbs' dividing surface; (ii) the radius of gyration of the interfacial electron is essentially the same as in the bulk; and (iii) the absorption spectral peak is the same whether at the interface or in the bulk, which has also been predicted in calculations (J. Phys. Chem. A 118, 7507 (2014)). Hence, the p-orbitals are likely to not to be distorted, but the blue tail may be different because this arises from excitation to more delocalised orbitals, as we comment on in the manuscript. We have also added a brief discussion of why our experiment is particularly sensitive to the asymmetry of the surface.

- For the blue side, the authors provide the following “or it may arise because the blue edge of the spectrum is associated with excitation to more diffuse orbitals”. For the bulk aqueous electron, the asymmetric absorption tail into the blue is associated with transitions to the conduction band of water contributing at the shortest wavelengths. For this data, it would

seem not only is the peak narrowed on the blue side, but that there is no blue-most 'tail' either; the absorption simply plummets rapidly at 600 nm. This is interesting as it could indicate that the conduction band is fundamentally altered in an interfacial experiment.

This is a nice point. We perhaps did not want to go quite as far as this, but on reflection, this is a nice comment that we have included in the manuscript.

Reviewer #5 (Remarks to the Author):

Great initiative – thank you for your contribution.

Reviewers' Comments:

Reviewer #4:

Remarks to the Author:

The authors have revised the manuscript in line with all comments and concerns. This paper is suitable for publication in Nature Comm.

In response to reviewer comments:

REVIEWERS' COMMENTS

Reviewer #4 (Remarks to the Author):

The authors have revised the manuscript in line with all comments and concerns. This paper is suitable for publication in Nature Comm.

There is no response required.

Best wishes

Jan